# Human Embryonic-Derived Mesenchymal Progenitor Cells (hES-MP Cells) are Fully Supported in Culture with Human Platelet Lysates

**DOI:** 10.3390/bioengineering7030075

**Published:** 2020-07-20

**Authors:** Sandra M. Jonsdottir-Buch, Kristbjorg Gunnarsdottir, Olafur E. Sigurjonsson

**Affiliations:** 1The Blood Bank, Landspitali—The National University Hospital of Iceland, Snorrabraut 60, 101 Reykjavik, Iceland; sandra@platome.com (S.M.J.-B.); oes@ru.is (K.G.); 2Faculty of Medicine, University of Iceland, Vatnsmyrarvegur 16, 101 Reykjavik, Iceland; 3Platome Biotechnology, Alfaskeid 27, 220 Hafnarfjordur, Iceland; 4School of Science and Engineering, University of Reykjavik, Menntavegur 1, 101 Reykjavik, Iceland

**Keywords:** embryonic stem cells, mesenchymal stromal cells, blood platelets, cell culture techniques, progenitor cells

## Abstract

Human embryonic stem cell-derived mesenchymal progenitor (hES-MP) cells are mesenchymal-like cells, derived from human embryonic stem cells without the aid of feeder cells. They have been suggested as a potential alternative to mesenchymal stromal cells (MSCs) in regenerative medicine due to their mesenchymal-like proliferation and differentiation characteristics. Cells and cell products intended for regenerative medicine in humans should be derived, expanded and differentiated using conditions free of animal-derived products to minimize risk of animal-transmitted disease and immune reactions to foreign proteins. Human platelets are rich in growth factors needed for cell culture and have been used successfully as an animal serum replacement for MSC expansion and differentiation. In this study, we compared the proliferation of hES-MP cells and MSCs; the hES-MP cell growth was sustained for longer than that of MSCs. Growth factors, gene expression, and surface marker expression in hES-MP cells cultured with either human platelet lysate (hPL) or fetal bovine serum (FBS) supplementation were compared, along with differentiation to osteogenic and chondrogenic lineages. Despite some differences between hES-MP cells grown in hPL- and FBS-supplemented media, hPL was found to be a suitable replacement for FBS. In this paper, we demonstrate for the first time that hES-MP cells can be grown using platelet lysates from expired platelet concentrates (hPL).

## 1. Introduction

The field of regenerative medicine has rapidly expanded in recent years. Various types of somatic stem cells, as well as embryonic and induced pluripotent stem cells, have been evaluated for their regenerative abilities [1,2,3,4].

Adult human mesenchymal stromal cells (MSCs) have been studied extensively and are considered promising candidates for regenerative therapies due to their differentiation potential and immunomodulatory function [5,6,7]. Several limitations are associated with MSCs that may hamper their use in regenerative medicine. These include inter-donor variability, variable isolation protocols, which commonly result in a heterogeneous population of cells, reduced proliferation capacity with cell age, and impairment of biological characteristics following long-term in vitro expansion [5,8,9,10,11].

Embryonic stem cells (ESCs) have the potential to provide a uniform and unlimited source of cells with efficient long-term proliferation and differentiation potential [3]. However, ESCs can cause teratoma formation, which reduces their applicability [1]. Various protocols are available that describe the derivation of cells with high MSC resemblance from ESCs [12,13,14,15,16,17,18], with the aim of combining the regenerative potential of MSCs with the long-term proliferation of ESCs. ESCs are commonly grown on feeder cells, such as mouse fibroblast cells [19]. Animal-free conditions are, however, recommended when deriving and growing cells intended for human use [1,20,21].

Human embryonic stem cell-derived mesenchymal progenitor (hES-MP) cells resemble MSCs and are derived from human ESCs under feeder-free conditions [22]. They demonstrate efficient long-term proliferation and can differentiate towards osteogenic, chondrogenic, and adipogenic lineages [23]. In addition, they do not form teratomas [23]. For those reasons, they have been suggested as a potential off-the-shelf product to replace MSCs [22,24].

In an effort to provide xeno-free conditions for culturing human cells, it has previously been demonstrated that human platelet lysate (hPL) supplementation is suitable as an animal serum alternative for MSC cultures [25,26,27,28]. Platelet lysates can be manufactured from expired human platelet concentrates (PCs) obtained directly from an accredited blood bank; this ensures that the platelets used have been obtained from healthy blood donors and have passed appropriate quality control, mandatory for manufacturing and issuing of blood components [26]. Recycling expired PCs reduces biological waste and avoids competition with blood banks for platelet donors, who are already in high demand [29]. 

Since hES-MP cells resemble MSCs, we hypothesized that hPL might be a good culture supplement choice to ensure the continuation of an animal-free growth environment. The purpose of this study, therefore, was to determine whether hPL is a suitable replacement for fetal bovine serum (FBS) as a cell culture media supplement for the growth and differentiation of hES-MP cells. Indeed, we demonstrated for the first time that hPLs produced from expired PCs are suitable as culture supplements for hES-MP cells. The lysates allowed hES-MP cells to maintain long-term proliferation, mesenchymal surface marker expression, and differentiation towards bone and cartilage.

## 2. Materials and Methods 

### 2.1. Ethical Statement

The study was approved by the National Bioethics Committee (number: VSN19-189).

### 2.2. Preparation and Characterization of hPLs

Ten expired PCs were acquired from the Blood Bank (Reykjavík, Iceland) and frozen (−80 °C) within 24 h after their expiry. Seven PCs were collected by apheresis (donor number, dn = 7), and three PCs were prepared from buffy coats (dn = 15). All PCs were obtained from healthy blood donors of the Blood Bank (22 males and 9 females, aged 18 to 64) and had undergone standard quality control.

Each PC underwent three freeze-thaw cycles (from −80 to 37 °C) to promote platelet lysis. The resulting lysate was platelet-depleted by centrifugation at 5000× *g* for 20 min. After centrifugation, the supernatant was removed and subjected to a second depletion step. Platelet fragments, visible as a pellet after each centrifugation step, were discarded. The supernatant was filtered through a 0.45 µm filter (Millipore, Billerica, MA, USA), and 40 IU/mL of heparin (Leo Pharma A/S, Ballerup, Denmark) was added. The resulting hPL was aliquoted and stored at −20 °C. Five batches of pooled platelet lysates were prepared. Three batches contained lysate from a buffy coat PC and an apheresis PC, while two batches were made from apheresis PCs only.

The human serum albumin concentration of hPL was evaluated with a Human Albumin ELISA Quantitation Kit (Bethyl Laboratories, Montgomery, TX, USA) to assess variability between the five batches prepared above (Appendix A); no significant batch variability was detected.

Growth factors were measured in both hPL (n = 5, for five hPL batches) and FBS (Gibco, Grand Island, NY, USA; n = 3). The concentrations of bone morphogenic protein 2 (BMP-2), basic fibroblast growth factor (bFGF), vascular endothelial growth factor (VEGF), insulin-like growth factor (IGF), and platelet-derived growth factor BB (PDGF-BB) were evaluated with a standard ELISA development kit (PeproTech, Rocky Hill, NJ, USA), and the transforming growth factor beta (TGF-β) concentration was evaluated with a Human TGF-beta1 Quantikine ELISA Kit (R&D Systems, Minneapolis, MN, USA). The concentrations of growth factors found in hPL were compared to the concentrations found in FBS (Appendix A).

### 2.3. Cell Culture and Proliferation

Human bone marrow-derived MSCs from three human donors were acquired from Lonza (Walkersville, MD, USA), and hES-MP cells (hES-MP002.5) were donated by Takara Bio Europe AB (previously Cellartis AB), Gothenburg, Sweden [22]. The MSCs used here have been previously used and studied by our group [25,26], and they adhere to International Society for Cellular Therapy (ISCT) criteria, as guaranteed by the manufacturer (the MSCs adhere to plastic under standard culture conditions and can be differentiated to adipogenic, chondrogenic, and osteogenic lineages. The ISCT standards regarding cell surface marker expression are followed). The hES-MP cells (from the same batch reported on in [22]) have previously been shown to differentiate to adipogenic, chondrogenic, and osteogenic lineages and to express several markers of MSCs [22], although they do not expand well on plastic and a gelatin layer must be used (as described below). The representative images of hES-MP cell morphology and of hES-MP cells differentiated into adipogenic, chondrogenic, and osteogenic lineages are provided in Appendix A.

MSCs and hES-MP cells were grown in DMEM/F12+ Glutamax medium (Gibco, Grand Island, NY, USA), 1% penicillin/streptomycin (Gibco), and either 10% hPL (hPL–hES-MP and hPL–MSC treatments/cells) or 10% FBS (FBS–hES-MP and FBS–MSC treatments/cells). The decision to use 10% hPL was made after evaluating different hPL concentrations; our group typically uses 10% hPL in studies with bone marrow-derived MSCs, and this concentration has been investigated in other studies [30,31,32], which matches the typical concentration of FBS used for supplementation.

The culture surface was coated with 0.1% gelatin (Sigma-Aldrich, St. Louis, MO, USA) to allow hES-MP cell attachment. The cells were grown under standard culture conditions (37 °C, 5% CO_2_, and 95% humidity). The medium was changed every 2 to 3 days, and cell passaging was performed when the cells reached 80% to 90% growth confluence. The cells were used for experimentation before reaching passage 8, except when evaluating the long-term proliferation and surface marker expression (10 passages).

Cell proliferation was evaluated with a cell population-doubling (PD) assay over 10 passages for MSCs (n = 3) and hES-MP cells (n = 6). The cells were maintained at standard culture conditions, and the cell count was determined at the end of each passage using a Neubauer hemocytometer (Assistant, Munich, Germany). The number of PDs at each passage was then used to find the cumulative PDs (CPDs) using Equations (1) and (2), as previously described [28,32]: (1)$$PDs=\frac{\log_{10}\left(N_{H}\right)-\log_{10}\left(N_{0}\right)}{\log_{10}\left(2\right)},$$
(2)$$CPDs=\sum_{i=1}^{n}PDs_{i},$$
where *N*_0_ and *N_H_* represent the number of cells seeded and the number of cells harvested (at the end of the expansion period), respectively, and *n* represents the number of passages.

### 2.4. Surface Marker Expression

The surface marker expression of hES-MP cells was evaluated with flow cytometry after expansion in either 10% hPL (hPL–hES-MP) or 10% FBS (FBS–hES-MP) for 4, 6, and 10 passages. Cells were stained with CD10, CD13, CD29, CD44, CD45, CD73, CD105, CD184, and HLA-DR antibodies (BD Biosciences, San Jose, CA, USA) according to manufacturer’s instructions and then fixed with 0.5% paraformaldehyde (Sigma-Aldrich) in phosphate-buffered saline (PBS, Gibco). Appropriate isotype-matched controls were used as negative controls. Samples were measured with a FACSCalibur flow cytometer equipped with an argon ion laser (BD Biosciences) and subsequently analyzed using Cellquest Pro software (BD Biosciences). Cells were gated according to their side scatter and forward scatter profiles.

Note that at the time of this study, we were unable to test for CD90. However, in subsequent work (unpublished data), the same batches of hES-MP cells were confirmed to be positive for CD90.

### 2.5. Osteogenic Differentiation

The hES-MP cells were differentiated towards the osteogenic lineage following two passages in media supplemented with either FBS (FBS–hES-MP) or hPL (hPL–hES-MP). Three thousand hES-MP/cm^2^ were seeded and then cultured in hMSC Osteogenic Differentiation Medium BulletKit^®^ (Lonza) for 28 days under standard culture conditions. The differentiation media was fully changed every 2 to 3 days. The cells were harvested after 0, 7, 14, and 28 days of osteogenic differentiation for gene expression analysis, the analysis of alkaline phosphatase (ALP) activity, and the quantification of mineralization with alizarin red staining. The ALP activity was tested with an alkaline phosphatase activity assay (Sigma-Aldrich) and normalized to sample protein concentration using a BCA assay (Pierce Biotechnology, Rockford, IL, USA), as previously described [26]. Mineralization was visualized with alizarin red staining and then quantified after dissolving the bound dye with cetyl-pyridinium chloride (Sigma-Aldrich). The cultures were fixed with 10% formaldehyde (Sigma-Aldrich) and then washed with distilled H_2_O prior to being stained with alizarin red for 20 min. After staining, the cultures were washed four times with dH_2_O and allowed to dry. Before quantification, the cultures were rehydrated with dH_2_O overnight, and then the bound alizarin red dye was dissolved with 10% cetyl-pyridinium chloride for 15 min. Subsequently, the Optical Density (OD) at 562 nm was measured with a photospectrometer (Thermo Scientific, Vantaa, Finland).

### 2.6. Chondrogenic Differentiation

Chondrogenic differentiation was performed in pellet cultures. First, 2.5 × 10^5^ hES-MP cells were seeded in hMSC Chondrogenic Differentiation Media (BulletKit^®^, Lonza) in 1.5 mL microtubes (Sarstedt, Nümbrech, Germany). The tubes were then centrifuged at 150× *g* for 5 min to generate a cell pellet. The tube lids were then punctured with a 2 mm sterile needle (Misawa, Tokyo, Japan) to facilitate gas exchange. The medium was fully changed every 2 to 3 days. Chondrogenic pellets were harvested after 0, 7, 14, 28, and 35 days for both gene expression analysis and the evaluation of glycosaminoglycan (GAG) concentration. For GAG analysis, the pellets were washed with PBS three times and then digested for 3 h at 65 °C in a papain extraction reagent containing 0.1 M sodium acetate (Sigma-Aldrich), 0.01 M Na_2_EDTA (Carl Roth GmbH + Co. KG, Karlsruhe, Germany), 0.005 M cystein HCl (Sigma), and 8 µL of crystallized papain (Sigma-Aldrich). After papain digestion, the GAG concentration was evaluated with a Blyscan assay (Biocolor, Carrickfergus, UK) following the manufacturer’s instructions. The pellets were stained with hematoxylin and eosin staining and Masson’s trichrome staining after 28 days of differentiation.

### 2.7. Gene Expression Analysis

Osteogenic samples for gene expression analysis were harvested with trypsinization after 0, 7, 14, 21, and 28 days of differentiation, put in TRIzol^®^ Reagent (Thermo Scientific) and stored at −80 °C until RNA isolation was performed, which was done within two months. Chondrocytic pellets were harvested after 0, 7, 14, 28, and 35 days of differentiation, washed once in PBS, put in TRIzol^®^ Reagent and stored at −80 °C until RNA isolation within two months. Prior to RNA isolation, samples were homogenized in TRIzol^®^ Reagent using the gentleMACS™ Dissociator (Miltenyi Biotec, Bergisch Gladbach, Germany). RNA isolation was performed according to the TRIzol^®^ Reagent manufacturer’s instructions. After RNA isolation, RNA clean-up was performed using an RNeasy Plus Mini Kit (Qiagen, Hamburg, Germany), following the manufacturer’s instructions. Reverse transcription was performed using a GeneAmp^®^ RNA PCR kit (Applied Biosystems, Foster City, CA, USA), according to the manufacturer’s instructions. Real-time qPCR was performed on all samples with runt-related transcription factor 2 (*RUNX2*, Hs00231692_m1), secreted phosphoprotein 1 (*SPP1*, Hs00167093_m1), and SRY-box 9 (*SOX9*, Hs01001343_g1) as genes of interest. The two most stable reference genes were carefully chosen after the screening of several reference genes. TATA box-binding protein (*TBP*, Hs00427620_m1) and *YWHAZ* (Hs03044281_g1) were found to be most suitable and were used for the normalization of expression in all samples. TaqMan^®^ Gene Expression Assays (Applied Biosystems) were used in all experiments.

### 2.8. Statistics

GraphPad^®^ Prism version 5 (GraphPad Software, La Jolla, CA, USA) and Microsoft Excel 2013 were used to analyze all data except gene expression data. Two-way ANOVA was used to analyze statistical significance, followed by a Bonferroni correction as a multiple comparison correction. 

Relative gene expression was evaluated using REST-384 © version 2, followed by a pairwise fixed reallocation randomization test to determine statistical significance [33]. Two reference genes, *TBP* and *YWHAZ*, were used for normalization in each sample and relative values presented. Three independent donors of MSCs were used in all experiments. Experiments with hES-MP cells were performed in triplicate. Data are presented as mean ± standard error. A *p*-value of ≤0.05 was considered statistically significant.

## 3. Results

### 3.1. hES-MP Cells Growm in the hPL-Supplemented Medium Demonstrate Consistently High Proliferation Rates over Long Culture Periods

Proliferation was evaluated by counting PDs of MSCs and hES-MP cells at each passage for 10 passages (80 days) (Figure 1). The mean PDs for hES-MP cells were 2.63 ± 0.36 and 2.46 ± 0.374 PDs for FBS and hPL, respectively, whereas the mean PDs for MSCs were 1.69 ± 0.283 PDs for FBS and 0.974 ± 1.32 PDs for hPL. No significant differences in proliferation were noted based on the type of media supplement used. hES-MP cells and MSCs exhibited similar growth kinetics until passage 7 (P7), after which MSC proliferation continuously declined while hES-MP cell proliferation continued at a comparable rate to earlier passages (Figure 1). The difference in proliferation between cell types became significant at P7 (*p* ≤ 0.05) and remained significant at every passage thereafter, with the greatest difference observed at P10 (13.7 ± 0.89 CPDs; *p* ≤ 0.001).

### 3.2. hPL–hES-MP and FBS–hES-MP Cells Express CD10, CD13, and CD105 Differently

The surface marker expression in hES-MP cells was evaluated with flow cytometry after expansion in either FBS- or hPL-supplemented media (FBS–hES-MP or hPL–hES-MP, respectively) after P4, P6, and P10. The hES-MP cells strongly expressed CD29, CD44, and CD73, while CD45, CD184, and HLA-DR were expressed only at very low levels (Appendix A). The FBS–hES-MP cells did not express CD10 at P4, P6, and P10 (16.52 ± 0.29, 17.11 ± 0.48, and 15.72 ± 0.20 geometric mean fluorescence intensity (gMFI), respectively) (Figure 2A). The hPL–hES-MP cells, on the other hand, did express CD10. The CD10 expression in the hPL–hES-MP cells was strongest at P6 (*p* ≤ 0.05; n = 3), with a fluorescence intensity of 114.6 ± 8.54 gMFI. At P4 and P10, the CD10 expression values were 71.6 ± 0.71 and 61.3 ± 8.74 gMFI, respectively (Figure 2A). Both the FBS–hES-MP and hPL–hES-MP cells were CD13-positive (Figure 2B). However, the CD13 expression remained constant in the FBS–hES-MP cells but increased in the hPL–hES-MP cells over time (*p* ≤ 0.01 between P4 and P6; *p* ≤ 0.05 between P6 and P10, n = 3). The CD13 expression was significantly stronger in the hPL–hES-MP cells than in the FBS–hES-MP cells at both P6 and P10 (322.8 ± 4.60 and 491.0 ± 56.71 gMFI, *p* ≤ 0.001, n = 3). The hES-MP cells expressed CD105 in all cultures, but the expression was significantly lower in the hPL–hES-MP cells than in the FBS–hES-MP cells at P4 (*p* ≤ 0.01, n = 3) and P10 (*p* ≤ 0.001, n = 3), with the difference in expression being 165.9 ± 10.9 and 228.9 ± 10.0 gMFI, respectively (Figure 2C).

### 3.3. Alkaline Phosphatase and Glycosaminoglycans Levels Follow Defined Patterns during Differentiation after Culture in Media Supplemented with either hPL or FBS

The hPL supplementation of cells prior to differentiation was not found to affect the activity of ALP during osteogenic differentiation or the GAG levels during chondrogenesis (Figure 3). Following growth under standard conditions, the FBS–hES-MP and hPL–hES-MP cells were induced towards osteogenic and chondrogenic differentiation after receiving the same differentiation treatment. ALP activity during osteogenic differentiation was evaluated after 0 (control), 7, 14, and 28 days of differentiation (Figure 3A). In the FBS–hES-MP cells, the ALP activity increased from 2.46 ± 0.74 to 39.0 ± 6.10 nmol(p-nitrophenol)/min/mg protein, while in the hPL–hES-MP cells it increased from 5.64 ± 1.21 to 40.62 ± 5.24 nmol(p-nitrophenol)/min/mg protein. The GAG concentration was evaluated as ng of GAG per chondrocytic pellet (Figure 3B). The GAG concentration increased between days 7 and 14, from 249.6 ± 133.1 to 783.4 ± 423.9 ng/pellet in the FBS–hES-MP cells and from 41.9 ± 26.5 to 658.1 ± 359.1 ng/pellet in the hPL–hES-MP cells. After day 14, the GAG concentration gradually decreased. The histology further revealed hypertrophic cells and decrease in cell number at the pellet core by day 28 (Appendix A). No statistically significant differences were observed in ALP activities or GAG concentrations between cultures based on the choice of culture supplement used during expansion.

### 3.4. hPL–hES-MP Cells Mineralize Later than FBS–hES-MP Cells during Osteogenic Differentiation

The level of mineralization was evaluated after 0 (control), 7, 14, 21, and 28 days of osteogenic differentiation (Figure 4). An increase in mineralization was detected after day 21 for the FBS–hES-MP cells (1.21 ± 0.15 OD; *p* ≤ 0.001) and day 28 for the hPL–hES-MP cells (0.96 ± 0.12 OD; *p* ≤ 0.001). Mineralization increased significantly between days 21 and 28 in all cultures, with an increase of 0.79 ± 0.12 OD (*p* ≤ 0.01) for the FBS–hES-MP cells and 0.90 ± 0.12 OD (*p* ≤ 0.001) for the hPL–hES-MP cells. Mineralization was significantly greater in the FBS–hES-MP cells than in the hPL–hES-MP cells at days 21 and 28, with the observed difference being 1.15 ± 0.07 OD (*p* ≤ 0.001) and 1.04 ± 0.11 OD (*p* ≤ 0.001) on days 21 and 28, respectively (Figure 4).

### 3.5. Lineage-Associated Gene Expression is Observed during Differentiation following Culturing in the hPL-Supplemented Medium

The expression of *RUNX2* and *SPP1* was evaluated after 7, 14, 21, and 28 days of osteogenic differentiation, and *SOX9* was evaluated after 7, 14, 28, and 35 days of chondrogenic differentiation (Figure 5). The expression was compared to those of undifferentiated FBS–hES-MP and hPL–hES-MP cells. The hPL–hES-MP cells had the significantly higher expression of both *RUNX2* and *SPP1* during the differentiation phase than the FBS–hES-MP cells. The *RUNX2* expression in both FBS–hES-MP and hPL–hES-MP osteoblasts followed a similar pattern but was consistently higher in cells grown with hPL (Figure 5A). The differences ranged from 0.37 ± 0.04 fold (*p* ≤ 0.05; n = 3) to 0.53 ± 0.07 fold expression (*p* ≤ 0.01; n = 3). The *SPP1* expression in the FBS–hES-MP osteoblasts also followed a similar pattern to that in the hPL–hES-MP osteoblasts, except at day 21, where the *SPP1* expression increased in the hPL–hES-MP cells but continued to decrease in the FBS–hES-MP cells (Figure 5B). The expression of *SPP1* was consistently higher in the hPL–hES-MP cells than in the FBS–hES-MP cells, with the greatest difference observed at day 21 (0.59 ± 0.03 fold; *p* ≤ 0.01; n = 3).

The expression of *SOX9* increased with time in chondrocytes derived from both the FBS–hES-MP and hPL–hES-MP expansion cultures until day 28, after which a slight decrease was observed (Figure 5C). The *SOX9* expression was observed to be higher in hPL chondrocytes at earlier time points, but at day 28, a 4.27 ± 0.04 fold greater increase was observed in the *SOX9* expression in the FBS chondrocytes than in the hPL chondrocytes, after which the expression remained higher in FBS chondrocytes (by 3.18 ± 0.07 fold on day 28) and 3.97 ± 0.08 fold on day 35; *p* ≤ 0.001; n = 3).

## 4. Discussion

hES-MP cells have been reported to have similar characteristics and bioactivity to MSCs [22]. Due to this resemblance, they have been suggested as a potential off-the-shelf product that can be handled and used in an identical manner to MSCs [22,34]. hPL has been widely described as a promising culture supplement for MSCs, but the evidence regarding hES-MP cells has been lacking to date [35]. The goal of the current study was to evaluate, for the first time, how maintaining hES-MP cells in an hPL-supplemented culture medium from expired platelets affects their in vitro growth and differentiation.

When the culture medium was supplemented with hPL, the hES-MP cells demonstrated excellent long-term growth and exhibited the expression of key surface markers previously described for MSCs [36,37]. The evaluation of proliferation over 10 passages revealed that the hES-MP cells exhibited a continuous rate of proliferation for a longer time than the MSCs, with no signs of senescence at the end of the experiment (Figure 1). The MSC growth, on the other hand, began to decline after five passages. No differences in proliferation were found based on whether the hES-MP cells were grown in FBS- or hPL-supplemented media.

The expression of cell surface markers commonly seen on the MSCs was evaluated over 10 passages and compared between treatments (Figure 2). The CD10 expression was detected on the hES-MP cells from hPL-supplemented culture media, but not in FBS-supplemented culture media. The Expression of CD10 on MSCs has been a subject of controversy for a long time, with previous studies reporting both positive and negative expression [38,39]. Karlsson et al. [22] reported that hES-MP cells express CD10 when supplemented with a human serum. A review of the literature revealed several studies that evaluated cell surface markers expressed on hES-MP cells following expansion in FBS-supplemented media, none of which reported on the expression of CD10 [23,24,34,40,41]. CD10 has been identified as playing a role in maintaining the stem cell pool in various body compartments, especially the early progenitor population. This is achieved by its endopeptidase activity, which cleaves proteins in the cellular microenvironment and thus prevents them from promoting differentiation [42,43]. The expression of CD10 is associated with higher stemness of expressing cells when compared to nonexpressing cells [42]. Here, we demonstrated for the first time that hES-MP cells do not express CD10 after expansion in FBS-supplemented media, which possibly points to lower overall stemness of the hES-MP population following culture media supplementation with FBS. Further studies into factors causing CD10 expression, or lack thereof, based on the expansion media used are warranted.

The expression of both CD13 and CD105 was detected, regardless of whether FBS or hPL was used as a culture media supplement (Figure 2). The expression of CD13, however, was found to increase with time for the hPL–hES-MP cells. The CD105 expression was significantly lower in hPL–hES-MP cells than in the FBS–hES-MP cells. CD13 (also known as aminopeptidase N) has been described as a cellular marker for MSCs and other adult stem cells, even though its role in stem cell biology remains largely unknown [44]. Studies done on liver cancer stem cells and skeletal muscle satellite stem cells have demonstrated the role of CD13 in the survival of the stem cell pool [44,45]. Gabrilovac et al. [46] demonstrated that TGF-β increases the expression of CD13 on HL-60 cells in a concentration- and time-dependent manner. The time-dependent increase in CD13 expression seen in the hES-MP cells grown in the hPL-supplemented medium could be explained by the higher amount of TGF-β observed in hPL than in FBS (Appendix A). The significance of the CD13 expression in the hES-MP cells, however, remains unknown.

The stronger CD105 expression was associated with the hES-MP cells cultured in the medium supplemented with FBS. CD105 (also known as endoglin) is a co-receptor for ligands of the TGF-β family, such as TGF-β1 and BMP-2, and is important for activating the Smad signaling pathway [47,48]. The role of CD105 in cancers has been studied extensively and is commonly associated with migration and adhesion of cells, as well as being important for angiogenesis [47,49,50,51]. The role of CD105 in the cell biology of MSCs and hES-MP cells, however, has been studied only to a very limited extent and remains largely unknown [50,52].

It was found that the hES-MP cells successfully differentiated towards both osteoblasts and chondrocytes after expansion in culture media supplemented with hPL or with FBS. The ALP activity increased with time, indicating successful differentiation towards osteoblasts (Figure 3A). Tissue mineralization is normally seen at advanced stages of differentiation. The in vitro mineralization of MSCs is usually detected after 21 days of growth in the presence of osteogenic stimulants [53]. Our results for hES-MP cells are therefore consistent with previous findings for MSCs; in this study, hPL–hES-MP cells mineralized later (day 28) than FBS–hES-MP cells (day 21) (Figure 4). This may be related to the fact that hPL contains significantly higher levels of TGF-β than FBS. TGF-β has been described as an inhibitor of osteogenesis, promoting the development of pre-osteoblasts but inhibiting the development of mature osteoblasts and mineralization [54]. During differentiation, all hES-MP cell cultures were treated with the same osteogenic induction media, regardless of whether they were expanded in hPL or FBS, and thus high levels of TGF-β were not present at the time of differentiation. However, pre-treatment with high levels of TGF-β prior to differentiation might encourage the cells to mineralize later than when pre-treated with low levels of TGF-β.

The significantly higher expression of the osteogenic markers *RUNX2* and *SPP1* was observed in osteoblasts derived from hPL than in FBS-supplemented hES-MP cell cultures at all time points evaluated (Figure 5). The upregulation of osteogenic markers and the increased ALP activity during differentiation are both suggestive of successful osteoblast formation and indicate that growing hES-MP cells in hPL-enriched culture media allows them to preserve their osteogenic potential. How delayed mineralization will affect the quality of differentiation remains to be determined.

Chondrogenic differentiation was evaluated in terms of the accumulation of GAG and the expression of the chondrocytic marker *SOX9*. hES-MP cells cultured in the hPL-supplemented medium prior to differentiation initially expressed SOX9 at significantly higher levels than cells cultured in the FBS-supplemented medium (at 7 and 14 days), but this was reversed at later time points (Figure 5C). In all cultures, the *SOX9* expression peaked at day 28 and subsequently declined, similar to what has been reported by others [55]. However, at day 28, chondrocytes derived from hES-MP cells grown in the FBS-supplemented medium prior to differentiation had four-fold expression of *SOX9* compared to those derived from cells from the hPL-supplemented medium. GAG concentrations were initially observed to increase, but they subsequently decreased over a period of five weeks. No differences were observed, based on which culture supplement the cells were treated with prior to differentiation (Figure 3B). These findings are consistent with previous reports for MSC chondrogenic differentiation [25].

It should be noted that there was a slight difference in processing the hPL and FBS media supplements in this study: hPL was subjected to centrifugation and filtration steps, while FBS was not. This difference could contribute to the differences observed between cells derived in FBS- and hPL-supplemented culture media.

In the current study, the presence of extracellular vesicles (EVs) in hPL was not considered. The presence of EVs in hPL, such as fibrinogen, can result in gel formation in culture media as well as the precipitation of fibrin during storage and cell culture, and it may influence cell migration, proliferation, and differentiation [56]; therefore, EVs in the culture media supplements used in this study may influence the results to some degree. Therefore, in future work, the removal of EVs from hPL should be carried out (e.g., using the protocol developed by Pachler et al. [56]) and confirmed (e.g., via the monitoring of the platelet marker CD41).

## 5. Conclusions

hPLs are suitable for enriching culture media for the growth of hES-MP cells and allow cells to maintain good growth, osteogenic and chondrogenic differentiation, and normal expression of cell surface markers. Since cells are derived under animal-free conditions, hPL can be used instead of FBS as a growth medium supplement to ensure the continuity of an animal-free environment during cell culture.

## Figures and Tables

**Figure 1 bioengineering-07-00075-f001:**
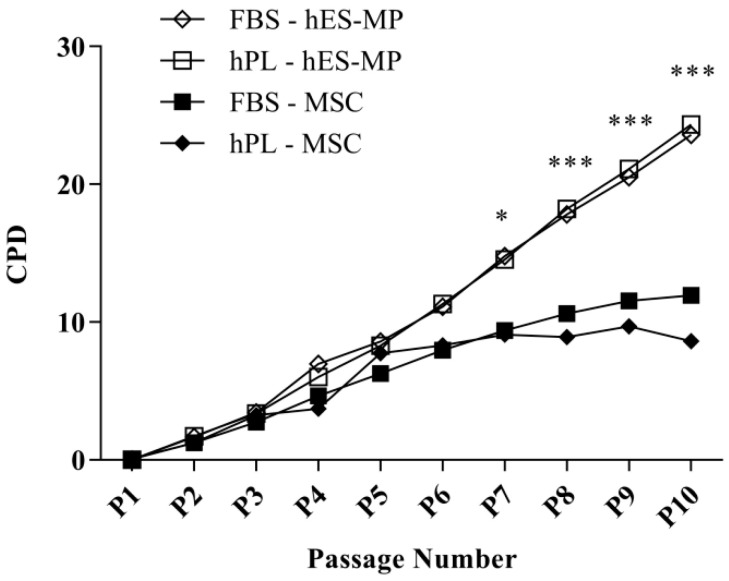
Long-term proliferation of human embryonic stem cell-derived mesenchymal progenitor (hES-MP) cells and mesenchymal stromal cells (MSCs). The MSCs experienced overall lower cumulative population doublings (CPDs) than the hES-MP cells. Differences in cell proliferation were not linked to the type of cell culture supplement used (human platelet lysate (hPL) or fetal bovine serum (FBS)). (* *p* ≤ 0.05; *** *p* ≤ 0.001).

**Figure 2 bioengineering-07-00075-f002:**
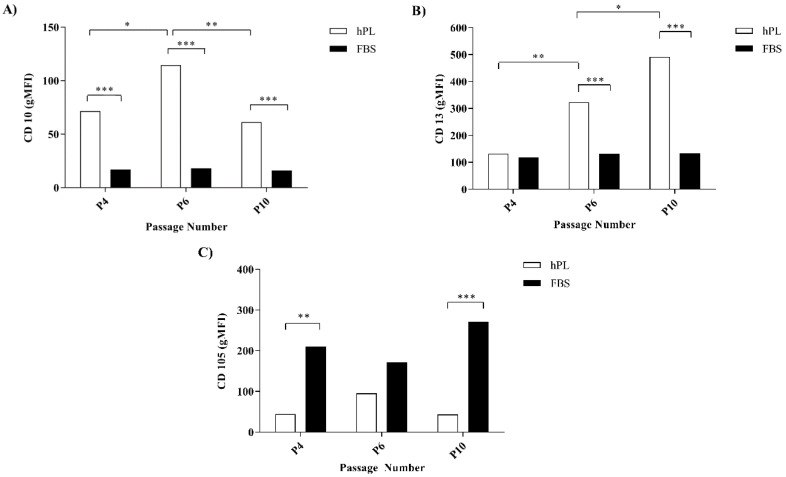
CD10 (**A**), CD13 (**B**), and CD105 (**C**) expression on the hES-MP cell surfaces over 10 passages. The CD10 expression was detected in the hPL–hES-MP cells at all passages, but not in the FBS–hES-MP cells. The CD13 expression was detected in all cultures with the hPL–hES-MP cells, demonstrating time-dependent increases, while the FBS–hES-MP cells showed constant expression. The CD105 expression was significantly lower in the hPL–hES-MP cells than in the FBS–hES-MP cells at all timepoints. (gMFI = geometric mean fluorescence intensity; * *p* ≤ 0.05; ** *p* ≤ 0.01; *** *p* ≤ 0.001).

**Figure 3 bioengineering-07-00075-f003:**
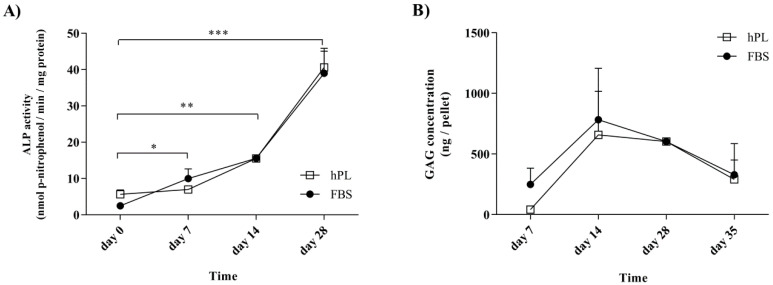
Lineage-specific bioactivity during osteogenic and chondrogenic differentiation. (**A**) During osteogenic differentiation, the alkaline phosphatase (ALP) activity significantly increased in a time-dependent manner over four weeks. (**B**) An initial increase was seen in GAG levels in chondrogenic cultures, with a subsequent decrease after two weeks of differentiation as chondrogenesis advanced. (* *p* ≤ 0.05; ** *p* ≤ 0.01; *** *p* ≤ 0.001).

**Figure 4 bioengineering-07-00075-f004:**
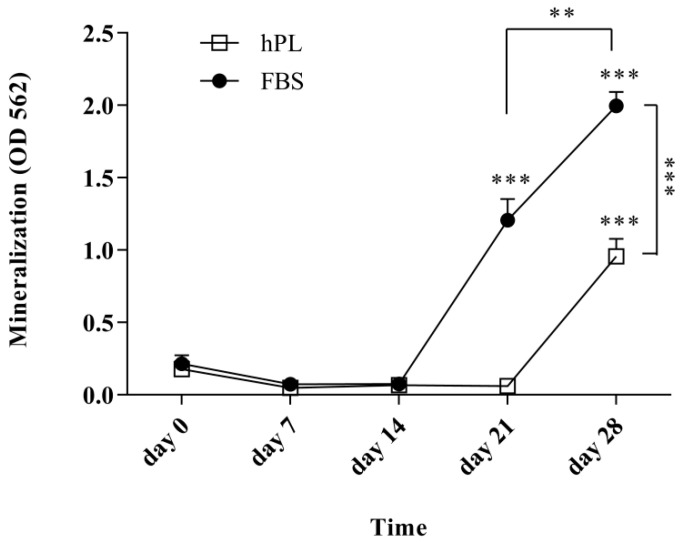
Mineralization during osteogenic differentiation. Significant mineralization was observed after 21 and 28 days for the FBS–hES-MP and hPL–hES-MP cells, respectively. The level of mineralization was significantly higher in the FBS–hES-MP cells than in the hPL–hES-MP cells by the end of 28 days. (** *p* ≤ 0.01; *** *p* ≤ 0.001; *p*-values directly above data points were compared with that data point on day 0.)

**Figure 5 bioengineering-07-00075-f005:**
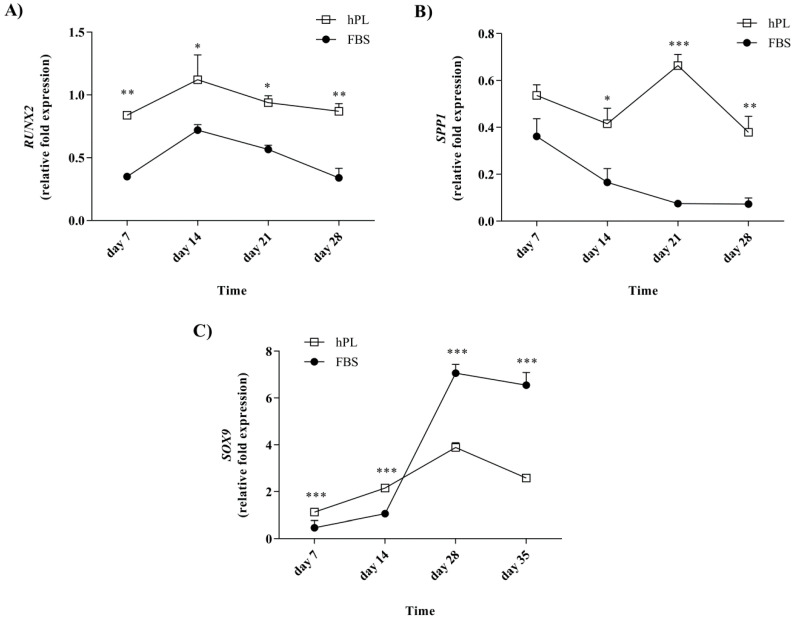
Lineage-specific gene expression during osteogenic and chondrogenic differentiation. During osteogenic differentiation, the hPL–hES-MP cells had the significantly higher expression of both *RUNX2* (**A**) and *SPP1* (**B**) than the FBS–hES-MP cells. (**C**) The *SOX9* expression increased with time in chondrocytes derived from both the FBS–hES-MP and hPL–hES-MP expansion cultures until day 28. The *SOX9* expression was initially higher in the hPL–hES-MP cells, but after day 14 the expression was higher in the FBS–hES-MP cells. (* *p* ≤ 0.05; ** *p* ≤ 0.01; *** *p* ≤ 0.001).

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
