# Peer review of "Human Embryonic-Derived Mesenchymal Progenitor Cells (hES-MP Cells) are Fully Supported in Culture with Human Platelet Lysates"

_bioengineering, 2020, doi:10.3390/bioengineering7030075_

Round 1

Reviewer 1 Report

The article “Human Embryonic-Derived Mesenchymal Progenitor Cells (hES-MP Cells) are Fully Supported in Culture with Human Platelet Lysates” aims to compare the proliferation and differentiation of mesenchymal progenitors derived from human embryonic stem cells (hES-MP) and mesenchymal stromal cells (MSC) expanded with culture medium supplemented with fetal bovine serum (FBS) or human platelet lysate (hPL).  Due to their excellent proliferative potential, pluripotent stem cells are considered as an alternative to overcome the limitations of MSC for regenerative medicine. The article is well written and although not very exhaustive, it contributes to the optimization of MSC (or iMSC) studies, required for the translation of cell-based therapeutics from the laboratory into clinic. In order to further improve the quality of the manuscript, few aspects should be addressed:

  • Authors should explain why they supplemented the medium with 10% hPL. Other percentages of supplementation are described in the literature. Especially since it is the first time that this culture medium supplement is being evaluated to expand hES-MSC.
  • According to ISCT guidelines, MSCS must be plastic adherent when kept under standard culture conditions; express the positive and negative MSC markers and retain the ability to differentiate into adipocytes, osteoblasts and chondrocytes under the standard differentiation conditions. Authors should perform audiogenic differentiation studies to fulfill the three criteria proposed by ISCT.
  • It is not mandatory, but it would be useful to add the formula to calculate the CPD.
  • Some sub-titles need to be rewritten. It is not the cells that are supplemented, but the culture medium (title 3.1). In the same way that the cells did not undergo any treatment with FBS or hPL (titles 3.3 and 3.5). Cells were grown in culture medium supplemented with FBS or hPL. Authors should revise this term “treatment” throughout the manuscript.
  • Subtitle 3.4 is missing.
  • Line 187: the population doublings for MSC culture with medium supplemented with hPL is correct?
  • In order to be consistent, the order of the columns in graph B of figure 2 must be the same as in graphs A and C.
  • Line 284: Cells were maintained in hPL-supplemented medium not in hPL alone. Please revise this detail throughout the manuscript.

Reviewer 2 Report

  1. In the Introduction, the authors state: “We have previously demonstrated the suitability of hPL for MSC cultures as an animal serum alternative [25,26]”, when actually it has been already demonstrated and reported before 2013 (for instance, Schallmoser et al. Human platelet lysate can replace fetal bovine serum for clinical‐scale expansion of functional mesenchymal stromal cells. Transfusion 2007, 47(8):1436-1446)
  2. In platelet units, many extracellular vesicles (EVs, represented by exosomes, microparticles and apoptotic bodies) released by platelets are present, which are not destroyed by the freezing and thawing cycle. The platelet lysate is first centrifuged at 5,000 g for 20 min and then filtered using 0.45 µm filters. Although the filtration step can successfully remove apoptotic bodies from the remaining supernatant, it does not remove larger and smaller extracellular vesicles. Indeed, a centrifugation step at 100,000 g or the use of 0.1 µm filters would remove EV-derived RNA that can have an impact on the cells (Pachler et al. A Good Manufacturing Practice–grade standard protocol for exclusively human mesenchymal stromal cell–derived extracellular vesicles. Cytotherapy 2017, 19:458–472). Furthermore, FBS was not subjected to such procedure, which could probably explain some differences observed between the 2 different growth conditions.
  3. With respect to surface markers, I think a point would have been to test CD41 (platelet marker) as a negative control, due to the interaction of cells with platelet-derived EVs.
  4. Why CD90 was not tested?
  5. Other groups who reported on the replacement of FBS with hPL, used lower percentages of hPL, such as 5 or 2.5% instead of 10%. Have the authors also tested different percentages?

Reviewer 3 Report

In this study Jonsdottir-Burch S. M. and collaborators demonstrate that hES-MP cells can be grown using platelet lysates from expired platelet concentrates (hPL). Since use of FBS and similar supplements raise concerns regarding transmission of possible zoonoses, this study proposes hPL as an optimal xeno-free alternative for preclinical and clinical evaluation. Considering the difficulties in obtaining blood from banks for platelet donors, who are already in high demand, recycling expired platelet concentrates have the advantage to overcome this issue and reduces biological waste.

Overall the paper is interesting and well written. However, the following issues should be addressed in order to improve this study for the readers of Bioengineering:

  1. The introduction should be improved and better clarify the aim of the study.
  2. Authors should assess the phenotype of hES-MP cells compare to MSC grown in hPL and FBS. Representative pictures of the morphology of the cells should be provided.
  3. Tri-lineage differentiation should be demonstrated. Besides osteo and chondro differentiation, authors should add also adipogenic differentiation. Moreover, representative picture of the cells differentiated should be added.
  4. In supplementary figure S3 authors should clarify why CD184, CD45 and HLA-DR are not expressed in hES-MP. From the graph they seems to be expressed, although less than the other markers. Please make this point clear and, if necessary, change the graph type to better show the data obtained.
  5. Line 322-324 is not clear and should be re-write.

Round 2

Reviewer 1 Report

Line 205 - "grown" instead of "growm"

Line 279 - "during cell diferentiation" instead of "during differentiation"

Reviewer 3 Report

I believe that the authors have added all the information required and the manuscript is now improved. I recommend the publication in Bioengineering.